# An Anatomical Template for the Normalization of Medical Images of Adult Human Hands

**DOI:** 10.3390/diagnostics13122010

**Published:** 2023-06-09

**Authors:** Jay Hegdé, Nicholas J. Tustison, William T. Parker, Fallon Branch, Nathan Yanasak, Lorie A. Stumpo

**Affiliations:** 1Department of Neuroscience and Regenerative Medicine, Medical College of Georgia, Augusta University, Augusta, GA 30912, USA; 2Department of Radiology and Medical Imaging, School of Medicine, University of Virginia, Charlottesville, VA 22903, USA; 3Department of Radiology and Imaging, Medical College of Georgia, Augusta University, Augusta, GA 30912, USA

**Keywords:** image registration, magnetic resonance imaging (MRI), musculoskeletal radiology, upper extremity, wrist

## Abstract

During medical image analysis, it is often useful to align (or ‘normalize’) a given image of a given body part to a representative standard (or ‘template’) of that body part. The impact that brain templates have had on the analysis of brain images highlights the importance of templates in general. However, templates for human hands do not exist. Image normalization is especially important for hand images because hands, by design, readily change shape during various tasks. Here we report the construction of an anatomical template for healthy adult human hands. To do this, we used 27 anatomically representative T1-weighted magnetic resonance (MR) images of either hand from 21 demographically representative healthy adult subjects (13 females and 8 males). We used the open-source, cross-platform ANTs (Advanced Normalization Tools) medical image analysis software framework, to preprocess the MR images. The template was constructed using the ANTs standard multivariate template construction workflow. The resulting template image preserved all the essential anatomical features of the hand, including all the individual bones, muscles, tendons, ligaments, as well as the main branches of the median nerve and radial, ulnar, and palmar metacarpal arteries. Furthermore, the image quality of the template was significantly higher than that of the underlying individual hand images as measured by two independent canonical metrics of image quality.

## 1. Introduction

Analysis of medical images often requires ‘apple-to-apple’ comparisons or measurements [1,2,3]. For instance, if we are interested in determining whether a given brain region is atrophying in a given patient, we must be able to directly compare the relevant brain images from two different time points, while minimizing trivial variations between the two images (see below). Similarly, if we are interested in determining if the given brain region in the patient is abnormally small, we must be able to compare the given brain region from the patient to a suitable reference or standard (e.g., the population average brain derived from healthy patients).

A fundamental obstacle to such apple-to-apple comparisons is image variability. Any two medical images of a given body part are rarely identical, even when they are from the same patient, and obtained using the same imaging modality. In addition to the obvious image variations in the body part across patients and across imaging modalities, images can also vary depending on the imaging session and imaging parameters, such as viewpoint, field of view, etc. [4].

An effective method of minimizing image variations is to digitally align, or register, two images of the same body part to each other, so that the mutually corresponding anatomical regions are likely to occur in the corresponding coordinates of the two images [5]. A generalized, and common, version of the image registration process is called normalization [1,2], wherein all individual images of the same given body part are registered to the same common reference image, or template, so that the registered individual images have one-to-one correspondence. Thus, apple-to-apple comparisons among medical images require image registration or normalization, which in turn require a suitable template.

It is worth noting that a template is meant solely as an *image analysis tool* to be used in image registration, and not as an *anatomical model* for the given body part. Specifically, it is not meant as a substrate for anatomical measurements or inferences, clinical or otherwise, although the medical images processed using a template can, with suitable validation, have great clinical utility [5,6]. 

The availability of templates, therefore, tends to fundamentally improve the analysis of the corresponding body part. A classic example of this is the revolution in the analysis of brain images that was historically set off by the publication of such templates of the Talairach and MNI (Montréal Neurological Institute) templates for the human brain (for a recent review, see [1]). More recently, anatomical templates of other body organs, such as lungs, have had similar salutary effects [2].

However, there is currently no anatomical template available for the analysis of hands. We use our hands as both motor and sensory organs for performing countless daily tasks [7,8,9,10,11]. Even slight injury to, or pain in, the hand can adversely affect our day-to-day lives and degrade our sense of well-being [12,13]. In the U.S. and other developed countries, hand injuries are among the most common types of injury [14,15,16,17]. It is not surprising, therefore, that hand imaging is one of the most commonly performed medical imaging procedures [18]. This is because clinically treating hand-related clinical complaints typically requires imaging of the affected hand/s [14,15,16,17]. Together, these considerations indicate that the lack of a hand template is a significant barrier to progress in the treatment of hand-related clinical complaints.

To help overcome this barrier, we developed a publicly available anatomical template for the normalization of healthy adult human hands. In this brief report, we focus on the template for *healthy* adult hands. We constructed the template using 27 different anatomically representative hand images from 21 demographically representative healthy adults (13 females and 8 males; see Table 1 and Table 2 for additional demographic data).

The rationale for excluding children’s hands from this template was that children’s hands are significantly different from an anatomical viewpoint [19]. Similarly, the rationale for excluding diseased adult hands was to exclude the potentially confounding image variations in the diseased hands (see Discussion for additional information).

## 2. Materials and Methods

### 2.1. Subjects

A total of 37 adult subjects with healthy or diseased hands participated in this study. All were adult volunteers over 18 years of age. All subjects gave written informed consent prior to participating in the study. All study procedures were approved in advance by the Institutional Review Board (IRB) of Augusta University, where this study was carried out.

Of the aforementioned 37 subjects, 3 subjects had received a formal diagnosis of arthritis in both hands. The present report excludes the data from these three subjects with diseased hands and includes only the data from 34 subjects in which both hands were clinically healthy.

### 2.2. Hand Imaging

We carried out T1-weighted magnetic resonance imaging (MRI) of both hands in each of the 34 subjects. Each hand, including the wrist and the distal ≥ 5 cm of the forearm, was imaged in its natural resting position from the standard anteroposterior viewpoint using a 64-channel radiofrequency coil in a Siemens Magnetom Vida 3T scanner (voxel volume: 0.5 × 0.5 × 0.5 mm^3^). Each T1 image underwent two quality control steps. First, each image was examined for quality during the scan by a qualified MR technologist to ensure that the MR image was of clinical quality (i.e., it had the same resolution and met the other image quality requirements as the hand images obtained during standard clinical care of patients at our medical center). Second, each image was scrutinized clinically by a board-certified musculoskeletal radiologist who also screened the image against clinical standards. All of the 68 images passed both of these quality-control steps.

The resulting DICOM format images were converted to NIfTI (NII) format images using custom-written Matlab scripts (Mathworks, Natick, MA, USA). Each resulting three-dimensional (3D) NII image was 336 × 144 × 480 in size.

### 2.3. Template Creation Workflow

Figure 1 summarizes the template creation workflow. Briefly, of the resulting 68 T1 images of 68 different hands, 27 images were anatomically representative and had excellent image quality, as collectively judged by the authors, who included two board-certified, practicing musculoskeletal radiologists (WTP and LAS) and three medical image experts (JH, NJT, and NY). These 27 images were selected for creating the template. These came from 21 different subjects with fairly diverse demographic characteristics (Table 1 and Table 2).

**Listing 1.** Linux shell command used for running the ANTs template creation script noted in Figure 1. This script, <antsMultivariateTemplateConstruction.sh>, is publicly available at https://github.com/stnava/ANTs/blob/master/Scripts/, accessed on 1 June 2023. Each indented line represents a command line argument. See the script for an explanation of the arguments. The argument <inputHandImages> contains the list of the 27 NIfTI images of the hand from which the hand template was constructed. See text for additional details.antsMultivariateTemplateConstruction2.sh \-d 3 \-o ./Template/T_ \-r 0 \-n 0 \-w 1 \-k 1 \-q 100 × 70 × 0 × 0 \-f 6 × 4 × 2 × 1 \-s 2 × 1 × 0 × 0vox \-g 0.25 \-i 4 \-c 2 \-j 8 \-m CC [4] \-t BSplineSyN \-y 0 \$inputHandImages

The rationale for the above process of selecting the 27 source images for the template is this: It is well known that anatomical outliers or low-quality images reduce the quality of the template [20,21,22,23]. This is because the template creation process involves digitally registering the input images with each other. If the images are too dissimilar, the resulting template will be blurry. The greater the variance of the input images, the blurrier the template image [22,24]. Therefore, the input images must be somehow selected to minimize this variance.

When one is building a template of a given body part for the very first time, such as in this case, one has no choice but to select the input images more or less subjectively as described above. This is because from a computational perspective, template creation faces a classical bootstrapping (or ‘catch-22′) problem of self-starting processes: On the one hand, the input images have to be mutually similar in order to produce a high-quality template. On the other hand, making the images mutually similar using digital means would require a template. For this reason, it is common practice to use subjectively selected images for constructing the very first template of a given body part [20,21,22,23]. Since the hand template presented in this report is the very first template, the source images had to be subjectively selected as described above. Note that this initial template can be leveraged to build the next generation of hand templates using more objective image-selection methods (see Discussion).

We used the open-source, cross-platform ANTs (Advanced Normalization Tools) [2] medical image analysis software framework to carry out the preprocessing of the MR images and to construct the template [2]. Briefly, ANTsX is collection of advanced software tools based on the Python, R (and for some purposes, bash) scripting languages specifically designed for the processing of medical images [2]. The ANTsX ecosystem consists of a large collection of both advanced artificial intelligence (AI) and deep neural network (DNN) methods [2,25].

Of the aforementioned 27 images, 14 were images of the left hand. All of these 14 left-hand images were mirror-reflected about the vertical axis, so that they had the same orientation as the right hands. The resulting 27 ‘right-hand’ images were used as input to the ANTs standard bash shell script for multivariate template construction (https://github.com/stnava/ANTs/blob/master/Scripts/antsMultivariateTemplateConstruction.sh, accessed on 1 June 2023). The actual bash shell command line call to run the above script is shown in Listing 1. It is worth noting that our template-building software did not involve deep learning, or learning of any sort.

The above template-building script constructed the template in a stepwise manner. Briefly, images were first registered rigidly to minimize global variations in the orientation and positions of hands from one image to the next. This was followed by affine registration to remove further translational, rotational, scaling, and shearing differences. The final step consisted of symmetric normalization (SyN) deformable registration [22,26,27].

The post-processing of the raw template was carried out using standard procedures used for creating templates of other body parts, especially the brain [21,23,28]. To do this, we created a binary mask for the hand using k-means segmentation of the raw template using the ANTs function *kmeansSegmentation*(). We used this mask to set all the background voxels to zero. The resulting image was normalized to a bit depth of 256, so that individual voxels of the final template had values ranging from 0 to 65,536.

## 3. Results

### 3.1. Overview of the Hand Template

The resulting hand template is a 336 × 144 × 480 3D image (i.e., the same image dimensions as the aforementioned 27 input images from which the template was constructed). Note that, given the fact that the hand is inherently a 3D structure and that relaxed healthy hands naturally curve slightly toward the volar (i.e., palm) side, no single slice of the template captures the entirety of the hand.

Representative coronal slices of the template are shown in Figure 2. Similarly, representative sagittal and axial sections of the template are shown in Figure 3 and Figure 4, respectively.

It is apparent from a visual examination of the sections of the template that a substantial amount of anatomical details is preserved in the final template (also see Figure 5 below). Hyperintense regions corresponding to the various hand bones are apparent in the interior of the template, as are hyperintense regions corresponding to the skin. Regions with intermediate intensity corresponding to the various muscles, as well as hypointense regions corresponding to the tendons and blood vessels are also evident.

### 3.2. Physical Quality of the Template Image

Figure 5 shows an axial section of the template through the carpal region of the hand (Figure 5, *red horizontal line in the icon on the left*). This figure highlights the fact that some of the subtler anatomical structures, including nerves that can be hard to discern in individual T1 images, are clearly preserved in the template. Note, for instance, that the median nerve, as well as the superficial and deep branches of the ulnar nerve, is clearly visible (*arrows*). 

A detailed visual comparison of the template image with the 27 individual contributing hand images suggested the hypothesis that the physical quality of the template image was higher than the physical quality of the any of the individual images. We tested this hypothesis quantitatively using two standard metrics of image quality, structural similarity (SSIM) and mean-squared error (MSE) [29,30,31,32].

The distribution of SSIM values is shown in Figure 6A. An SSIM value of 1.0 indicates that a given image has the same quality as the template (*dashed blue arrow*), and lower values denote correspondingly poorer image quality. The quality of each of the 27 individual hand images was lower than that of the template, and this effect was statistically significant (one-tailed one-sample *t*-test, *t* = −2317.8; *df* = 26; *p* < 0.05).

Qualitatively similar results were independently obtained using MSE (Figure 6B). An MSE value of 0 denotes that a given image has the same physical quality as the template, and higher MSE values denote correspondingly poorer image quality. By this measure as well, the quality of the template image was better than of any of the individual hand images, and this effect was statistically significant (one-tailed, one-sample *t*-test, *t* = 617.5; *df* = 26; *p* < 0.05).

Together, these results demonstrate that the *physical* quality of the template image is significantly higher than the *physical* quality of any of the individual images from which it was constructed. Note that our analyses focused on the *physical* quality of the template image, and advisedly do not address the *perceptual* quality of the image because, as noted above, it is meant as an image analysis tool, as not as an anatomical model of the hand for clinical or other purposes.

## 4. Discussion

### 4.1. The Potential Utility and Impact of a Hand Template

This study presents, for the first time, an anatomical template for healthy adult human hands. Our results quantitatively demonstrate that our hand template is of high physical quality. We also show that our template preserves many of the clinically useful anatomical details.

Our template is a potentially highly useful tool in the analysis of hand images in two main respects: First, it represents an anatomical standard of the adult hand using which new, raw adult hand images can be normalized to a common coordinate frame. As noted above, normalization is a requisite initial step in making the aforementioned apple-to-apple comparisons. Therefore, our template will make such comparisons possible for hand images for the first time.

In general, high-quality templates in which the anatomical details are preserved, in turn help preserve these details in the normalized image [33,34,35]. In other words, high-quality normalization requires, among other things, a high-quality template. To the extent that our template preserves many of the anatomical details of the hand, it will help preserve the anatomical quality of the hand images during the normalization process.

A second respect in which our template is likely to be impactful in the analysis of hand images is to help build other hand templates, especially higher-quality hand templates or special-purpose hand templates (e.g., pediatric hand templates). This is because the availability of an initial template will allow users to digitally normalize the input images, thus obviating or minimizing the need for manually selecting the input images. In other words, the availability of our template will, once and for all, solve the aforementioned bootstrap problem for hand images.

### 4.2. Future Directions

As alluded to above, the template we present in this report is not meant to be a finished end product, but a useful starting point for future work. For instance, it will facilitate the construction of even more detailed, higher-resolution hand templates, akin to how initial templates of the human brain historically facilitated the construction of a diverse variety of better and more representative brain templates in humans and other species [19,21,36,37,38]. 

It is also worth noting that our template—or an initial anatomical template of any body part for that matter—represents a requisite step, but only an initial one, in the long bench-to-bedside translation pathway of scientific research to clinical applications.

One of the immediate goals of future work in the analysis of hand images is to develop normalization software that registers new hand images from multiple imaging modalities to our template. Fortunately, ANTs provides multiple publicly available software libraries in multiple computer languages that one can use to develop software that registers individual hand images to our template [2].

Our preliminary studies indicate that our template can indeed be used with this software to normalize novel T1 hand images, i.e., T1 hand images that played no part in constructing the template (unpublished results). Our results also indicate that our template is also fairly effective in normalizing images from other imaging modalities, such as fat-suppressed T2 images (not shown). A detailed evaluation of these registration results is beyond the purview of this preliminary report intended to publicly release our template, and will be addressed in a future publication.

It also remains to be established whether, and to what extent, our template is useful for the normalization of diseased hands or pediatric hands. Future work in this ongoing project will incorporate even greater anatomical detail into the template and seek to enhance the demographic representativeness of the template.

Our template can be freely downloaded from the GitHub repository at https://github.com/HegdeUSA/Hand_template, accessed on 1 June 2023. Also available for download at this repository are a 3D-printable model of the template and an anatomically segmented version of the template.

## Figures and Tables

**Figure 1 diagnostics-13-02010-f001:**
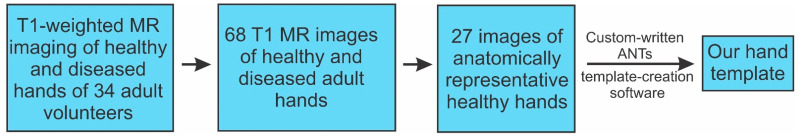
Hand-template creation workflow. See text for details about the workflow. For details about the ANTs software mentioned in this figure, please see Listing 1.

**Figure 2 diagnostics-13-02010-f002:**
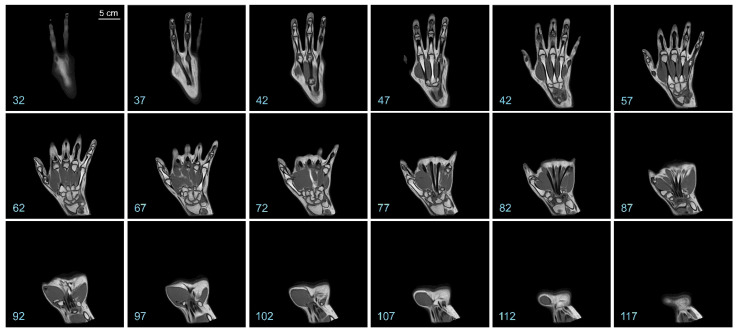
Representative coronal sections of the hand template. The *numbers at the bottom left* denote the slice numbers of the corresponding sectioning plane. The *scale bar* in the *top left panel* applies to all panels in this figure.

**Figure 3 diagnostics-13-02010-f003:**
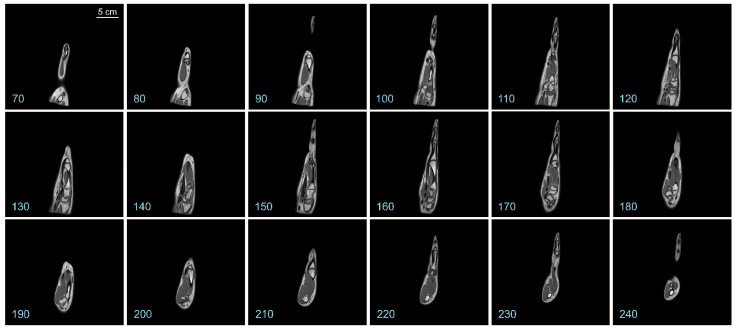
Representative sagittal sections of the hand template. The *numbers at the bottom left* denote the slice numbers of the corresponding sectioning plane. The *scale bar* in the *top left panel* applies to all panels in this figure.

**Figure 4 diagnostics-13-02010-f004:**
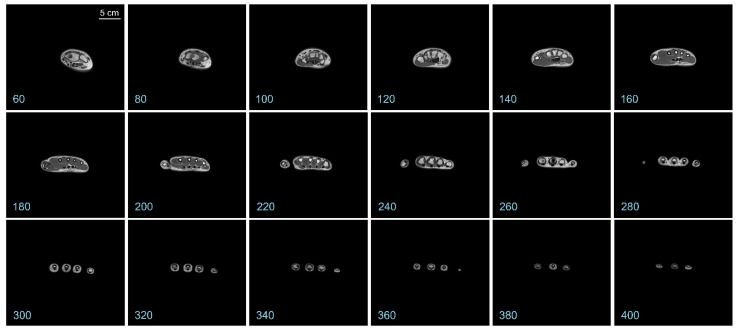
Representative axial sections of the hand template. The *numbers at the bottom left* denote the slice numbers of the corresponding sectioning plane. The *scale bar* in the *top left panel* applies to all panels in this figure.

**Figure 5 diagnostics-13-02010-f005:**
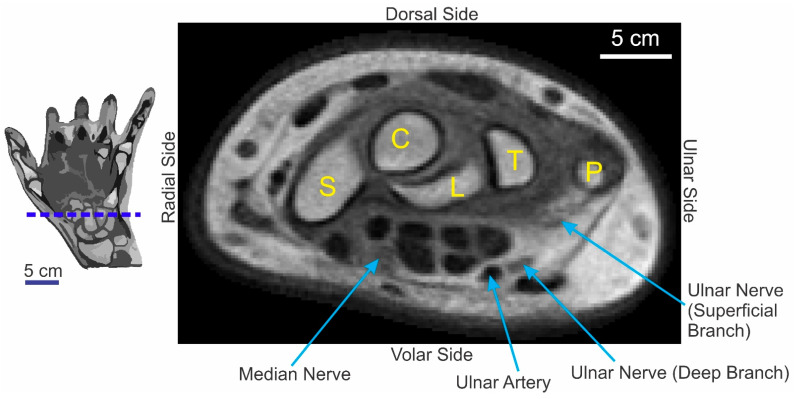
Some of the subtler anatomical features of the hand preserved in the template. This figure shows an axial section through the template. The plane of sectioning is denoted by the *dashed blue horizontal line* in the *icon on the left*. The *yellow letters* denote the carpal bones visible in the section. S, scaphoid; C, capitate; L, lunate; T, triquetrum; P, pisiform. *The blue arrows* denote some of the other anatomical structures. See text for additional details.

**Figure 6 diagnostics-13-02010-f006:**
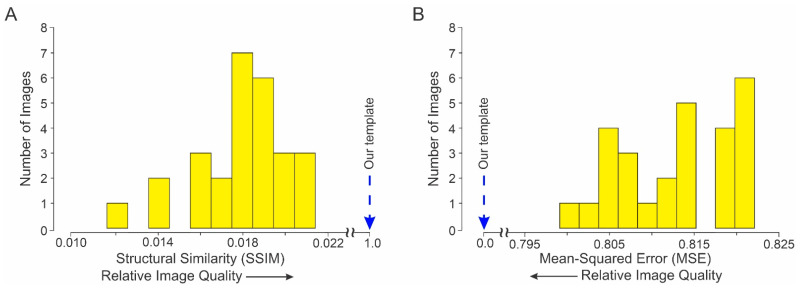
Comparison of the quality of individual hand images (*yellow histogram bars*) vs. the template (*dashed blue arrow*). The physical quality of each individual hand image was measured using two different standard metrices, structural similarity (SSIM) and mean-squared error (MSE). In each case, the template served as the requisite reference image. (**A**) Distribution of SSIM values. (**B**) Distribution of MSE values.

**Table 1 diagnostics-13-02010-t001:** Demographic characteristics of the 21 subjects whose hands contributed to the template.

Age	
Mean	35.9 years ± 17.9 (standard deviation)
Range	19.4–66.9 years
**Demographic Distribution**	
African American	2
Asian American	3
Caucasian	14
Latino	2

**Table 2 diagnostics-13-02010-t002:** Gender and handedness distribution of the 27 hand images that contributed to the template.

Subject Gender	Left Hand Only	Right Hand Only	Both Hands
Male	3	3	2
Female	5	4	4
Total	8	7	12
Grand Total			27 images

## Data Availability

The hand template described in this report is openly available from the GitHub repository at https://github.com/HegdeUSA/Hand_template. The underlying template creation software is openly available from the GitHub repository at https://github.com/stnava/ANTs/blob/master/Scripts/antsMultivariateTemplateConstruction.sh.

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
