# Peer review of "An Anatomical Template for the Normalization of Medical Images of Adult Human Hands"

_diagnostics, 2023, doi:10.3390/diagnostics13122010_

Round 1

Reviewer 1 Report

The authors explain that they focused on hand images and developed a template for healthy adult cases, after emphasizing the importance of templates in medical image analysis.

In order to contribute to the improvement of this brief report, I would like to raise some concerns as below.

2.3 Template Creation:

(1) What is the scientific/clinical basis for narrowing down the 68 T1 images to 27? If you can provide this basis, please let us describe in detail.

(2) Did you apply a neural network to generate the template image? If so, please briefly provide the architecture of the model. In addition, I understand that the main theme of this paper is "construction of a template image". Therefore, I think you should show a detailed flow chart of the “Custom written ANTS” part in figure 1. On the other hand, I feel that the other steps illustrated in figure 1 are unnecessary because they are sufficiently conveyed in sentences.

3.2 Quality of the Template Image:

(1) The process of quantitative analysis (application of a subjective evaluation, MSE, and SSIM) should be explained in the Methods section.

(2) Is "A detailed visual examination" an actual experiment by multiple observer? Or is it just your subjective feeling? This is confusing and should be properly explained.

(3) The results of described figure 6 are very difficult to understand.

Isn't it a natural result that SSIM and MSE to other images will be degraded if a template is applied to the reference?

SSIM and MSE are indices that easily degrade these values due to misalignment, etc., so I think it is a wrong use to apply them as quantitative indicators of image quality; These just show the similarity between the two images. Also, I did not understand the meaning of figure 6. For example, if the vertical axis is SSIM, the maximum value is 1.0, so I couldn’t make sure what you want to express.

4. Discussion:

(1) As pointed out above, your results do not quantify whether they are of high quality or clinically anatomical preservation. In order to state these claims, I think it is necessary to use indicators such as the mean opinion score (MOS), which is obtained by subjective evaluation by multiple doctors or medical professionals who are equivalent to doctors, instead of physical evaluations such as the SSIM.

(2) Please provide what kind of usefulness you can expect for the constructed template. This should be explained in relation to "lack of a hand template is a significant barrier to progress in the treatment" mentioned in the Introduction. That is, you should explain what barriers your template will allow you to overcome.

5. Conclusion:

I think that the purpose and conclusion should be matched and described appropriately. Please delete it if you don't need it.

Author Response

Please see the attached document named "Hegdé_etal_RESPONSES_TO_REVIEW_1.docx". Thank you.

Reviewer 2 Report

Medical records are poorly readable, no scale or markers (line 190-198). I suggest changing the form presentation and use a scale (markers) to clearly interpret the data pictorial. The image (line 201) is out of focus - all structures are visible, however, in case control and standard measurements - mainly with the use of rulers and very templates the quality of the photos is important - the higher the quality, the more accurate the calibration, which in turn translates into standard measurement tolerance. I propose to generate a new image with sharp edges of the structures and without pixelation.

The age group of patients on the basis of which the template was developed is very wide and for a reason very significant heterogeneous age differences. I propose to clearly describe in the text which ones algorithms were used in the development of the template and whether, for example, the patients taking part a experiment were pre-selected.

Please reconstruct tables 1a and 1b - the information contained in them can be presented compact - no need for a whole page. There is no specific data on what parameters were taken into account when creating templates - anthropometric and goniometric data, Hounsfield scale. Information - African American - 2, Asian American - 3, etc., does not constitute any added value, much less input data for scientific research. To be able to interpret the correctness of the construction and credibility of the obtained results, it is necessary to present specific input data for the development of templates

The article ended with tables - lack of a coherent summary binding the own study and summarizing the achievements of the authors. I suggest adding one constitutive point summary, experiment planning, process performed and results obtained. The final element is the development of the template, but the work submitted for review does not allow it for an unequivocal statement whether the final result has a rational justification from the point of view diagnostic.

Author Response

Please see the attached document named "Hegdé_etal_RESPONSES_TO_REVIEW_2.docx". Thank you.

Round 2

Reviewer 1 Report

I would like to thank the authors for their polite response to my comments.

I think that the present manuscript has mostly been appropriately revised and improved to be easy to understand.

However, just let me comment on the section describing the quality of the template images.

Regarding 3.2 (1) and (2) in the Author's Reply to the Review Report (Reviewer 1), I could not find the corresponding sentences in the revised manuscript. Please confirm.

Author Response

Please see the revised response. Thank you.
